# Effect of Wave, Current, and Lutocline on Sediment Resuspension in Yellow River Delta-Front

**Bowen Li** [1,2], **Yonggang Jia** [1,2,3,*], **J. Paul Liu** [4], **Xiaolei Liu** [1] and **Zhenhao Wang** [1,5]

[1] College of Environmental Science and Engineering, Ocean University of China, Qingdao 266100, China; l1398849394@163.com (B.L.); xiaolei@ouc.edu.cn (X.L.); wzh-ouc@foxmail.com (Z.W.)

[2] Shandong Provincial Key Laboratory of Marine Environment and Geological Engineering, Ocean University of China, Qingdao 266100, China

[3] Laboratory for Marine Geology, Qingdao National Laboratory for Marine Science and Technology, Qingdao 266100, China

[4] Department of Marine, Earth and Atmospheric Sciences, North Carolina State University, Raleigh, NC 27695, USA; jpliu@ncsu.edu

[5] First Institute of Oceanography, MNR, Qingdao 266061, China

* Correspondence: yonggang@ouc.edu.cn

**Abstract:** Historically, the Yellow River in China discharges $> 1 \times 10^9$ ton/yr sediment to the sea, and has formed a large delta in the western Bohai Sea. Its river mouth is characterized by an extremely high suspended sediment concentration (SSC), up to 50 g/L. However, the hydrodynamic factors controlling the high suspended sediments in the Yellow River estuary are not well understood. Here, we conducted two hydrodynamic observations and SSC measurements in the winter and spring low-flow seasons of 2014–2015 and 2016–2017 under five sea conditions, including calm-rippled, smooth-wavelet, slight, moderate, and rough, in the Yellow River Delta-front during the observation period. Under calm-rippled conditions, the contribution of currents to the total resuspended sediment concentration (RSC) was 77.7%–100.0%. During the smooth-wavelet and slight periods, the currents' contribution decreased as low as 30% and 3.0% of the total RSC, respectively. Under moderate and rough-sea conditions, waves accounted for at least 70% and 85% of the total RSC, respectively. The results indicate that 20 cm-thick lutoclines were created after a significant increase in the wave height to a peak value followed by a decrease. When the SSC is over 3 g/L and hydrodynamic conditions could not break the lutoclines, the flocculent settling of suspended sediment changes to hindered settling in the Yellow River Delta. Under hindered settling, the settling velocity decreases, and the resuspended sediments remains in the lutoclines and their lower water layers. This study reveals different controlling factors for the high SSC near a river-influenced delta, and helps us get a better understanding of a delta's resuspension and settling mechanisms.

**Keywords:** sediment resuspension; wave; current; lutoclines; in-situ observation

## 1. Introduction

Suspended sediments are an important part of sediment movement in estuarine coastal waters, and their distribution, diffusion, and deposition have a major impact on ports, waterways, and ecological environments. Additionally, marine sediments serve as a key sink for heavy metals that are released into the sea when the seabed sediment is resuspended [1,2]. Therefore, temporal changes in the suspended sediment concentration (SSC) are an important issue in estuarine coastal biogeochemical research. Previous studies have suggested that SSC variations in estuarine waters are significantly affected by flood and ebb tides, spring and neap tides, and seasonal factors [3–7]. A high SSC is the result of the combined effects of sediment transport and resuspension [8].

Suspended sediments are common in average flow and Stokes drift transport [9], and there is the high correlation between the coastal SSC and mean tidal current magnitude in tidal channels [10,11]. Tidal currents have greatly impacted upon suspended sediment distribution and transport, and SSC changes in the vertical profile are related to tidal runoff, which carries a large number of suspended sediments, resulting in an increase of SSC in the 0.5 m-thick layer of water above the sea bed [12,13].

In addition, the flux of sediment resuspension also controls the changes, source [14], and vertical distribution of suspended sediments concentration [15]. In estuaries, the sediment concentration variations are also related to seafloor sediment resuspension ([16]), which is the main source of suspended particulate matter in the water near the seabed [17,18]. Currents and waves are the main hydrodynamic factors inducing seabed sediment resuspension in estuarine water [19,20], particularly the shear stress they induce. When the current-induced shear stress exceeds the critical shear stress of the sediments, sediment particles separate from the seabed and resuspend in the water column [21]. Therefore, if the tidal current velocity is below the critical velocity, sediments are not resuspended and net sediment compaction occurs [22].

Previous studies show that, at shallow depths (<20 m), waves can directly act on the seabed and induce sediment resuspension [23–26]. Therefore, waves may become the main factor controlling sediment resuspension, and the RSC is correlated to the square of the wave orbital velocity [26,27]. However, the effects of waves on the seabed decrease with increasing water depth, and waves rarely cause seabed sediment resuspension at a large water depth (>80 m). For example, in the Gulf of Lion (NW Mediterranean), sediment resuspension on the inner continental shelf is mainly affected by waves in winter, but sediment resuspension on the southwest outer continental shelf is mainly controlled by bottom currents [28].

Some studies have been conducted on sediment resuspension and transport and their mechanisms, and some found that tidal currents and storms have a significant effect on SSC [29–34]. These studies were focused on the relationship between the resuspension mechanism and water depth. However, the suspended sediment concentration is the result of the collective influences of sediment transport, resuspension, and deposition, and the deposition rate is related to the SSC [35]. The deposition rate changes also affect sediment resuspension. This effect may be ignored when studying estuaries with a very low SSC; however, the SSC can exceed 4 g/L in the Yellow River Delta, therefore the deposition rate should be considered. Additionally, the resuspension mechanism is different under the different marine conditions in the Yellow River Delta [36], and storms are also an important factor inducing seabed erosion [37,38].

In this study, detailed field hydrodynamics and sediment data collected during the dry seasons, including the wave heights of a 50-year return period from 0 to 4 m [39], were used to quantitatively examine the formation and dynamics of suspended sediment in the Yellow River Delta. The SSC and sediment transport fluxes under different sea conditions were also analysed. Based on the importance of each physical process that contributes to the net sediment transport during the dry seasons, the sediment resuspension mechanisms in Yellow River Delta are further revealed.

## 2. Study Site

Historically the Yellow River contributes $1 \times 10^9$ ton/yr fluvial sediment to the ocean [40,41], and its delta has a high rate of deposition. The Yellow River derived sediment has been found being transported out and deposited all the way into the central Yellow Sea basin [42,43]. That the spatial distribution of suspended sediment in the Bohai Sea is mainly dominated by the river input and coastal resuspension [44]. The winter monsoon winds and strong waves are the main cause of the sediment resuspension and offshore transport [45]. The sediment shear strength is uniform under hydrodynamic action [46]. The waves are mainly induced by the wind in the Yellow River Delta and vary seasonally and interannually [47]. As the speed and frequency of the north-west winter monsoon is significantly higher than that in summer monsoon, in the winter the waves are strong [48] and the frequency of storm surges is much higher [49].

Previous studies show that the current velocity in the horizontal direction is closely associated with the water depth. The current velocity is the least when the water depth is less than 5 m, and the current velocity reaches the maximum at depths between 10 and 15 m and is reducing in depths from 15 to 20 m [50]. The area selected for in-situ measurement (38°10′ N, 118°54′ E) is located in the subaqueous delta front of the Yellow River (Figure 1). The water depths at the observation site range approximately from 8 to 12 m, and the current velocity is relatively high. The topography of the observation area inclines to the northeast at an average slope of 7–8°, and the gently sloped terrain is conducive to the formation of catastrophic storm surges [49]. For these reasons, the sedimentary processes are complex and the associated strata are disordered, exhibiting very high occurrences of resuspension in the research area [50,51].

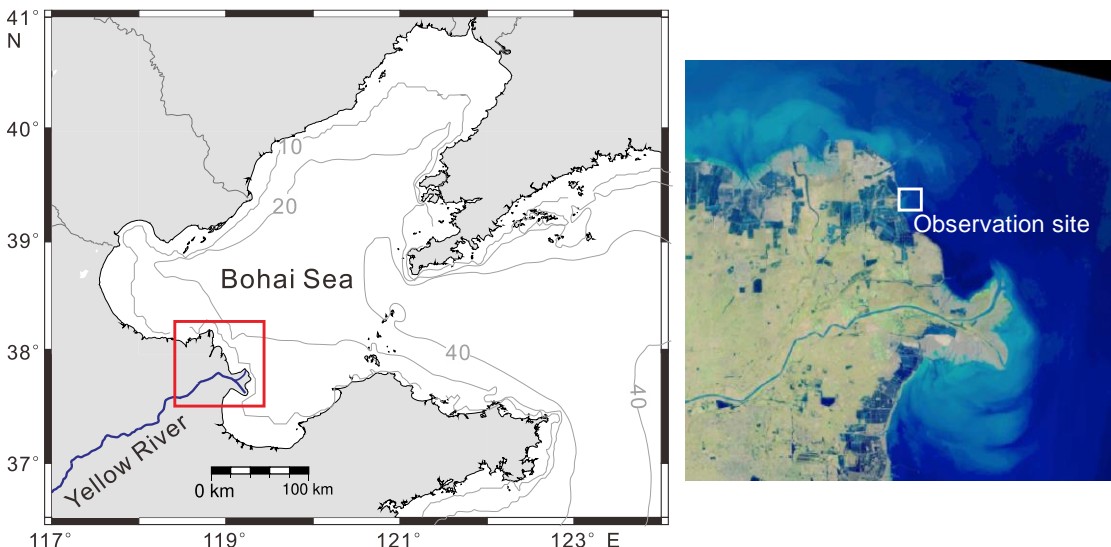

**Figure 1.** Location of the in-situ observation site in the Yellow River Delta, China.

## 3. Methods

### 3.1. Observation Instrumentation

A seabed-based tripod was designed to record the seabed surface hydrodynamic functions and SSCs. The design is optimized to maintain a stable posture on the seabed, and includes features to reduce subsidence and prevent the sideslip caused by storm waves and currents. The threaded holes on the tripod pedestals allow the installation of steel bars for insertion into the seabed. Therefore, the seabed-based tripod can remain stable, as the load is scattered through the pile foundations.

To measure erosion and deposition, current velocities, and turbidity we mounted a pack of sediment and hydrodynamic instruments on a tripod, including autonomous altimeter, current velocity meter, Argus surface meter, turbidity meter and wave tide gauge (see details in Figure 2). The range of the autonomous altimeter (AA400) is 0–3000 mm, with an instrumental error of 2%. The sampling frequency is 1.0 Hz, and the sampling interval is 30 min. The range, accuracy, and direction of the ALEC-EM current velocity meter is 0–500 cm/s, 1.0 cm/s, and eastward, respectively. Its sampling frequency and interval is 1.0 Hz and 10 min, respectively. The Argus surface meter (ASM-4) consists of tilt sensors, temperature sensors, microcontrollers, a memory stick, a power supply, and a sensor stick with a length of 96 cm and sensor spacing of 1 cm; the range is 0–2000 NTU, and the sampling interval is 15 min. The range and instrumental error of the XR-420 turbidity meter is 0–4000 NTU and 2%, respectively, while the sampling frequency and interval is 1 Hz and 2 min, respectively. The range and instrumental error of the wave tide gauge (RB16-TWR-2050) is 0–25 m and 0.05%, respectively. This gauge measures the water pressure every 30 min with a sampling frequency of 1 Hz. To assess the sediment strength, a 5 m sediment core was collected from the observation site.

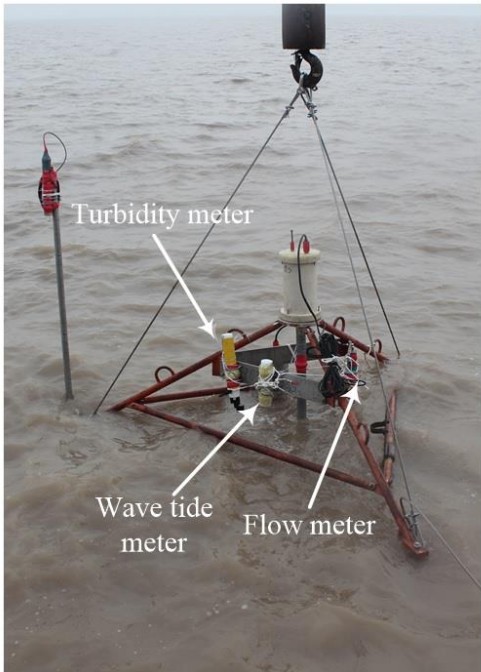 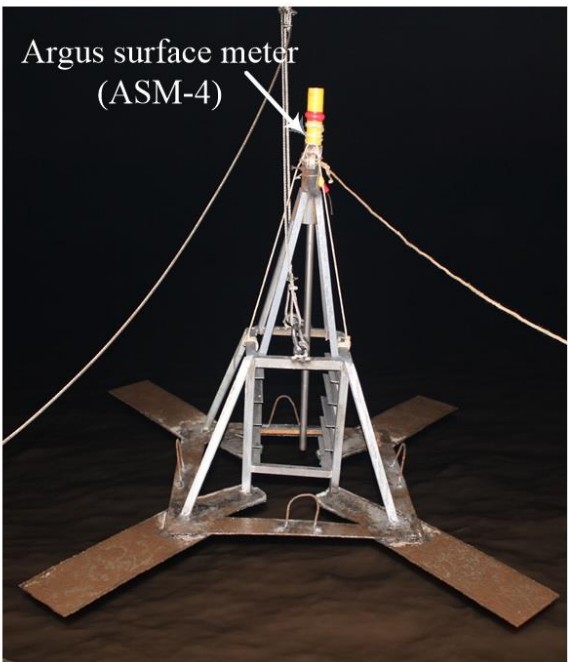

**Figure 2.** Observation system consisting of a wave–tide gauge (RB16-TWR-2050), ALEC-EM current-velocity meter, XR-420 turbidity meter, and ASM-4.

### 3.2. Performance of Observation Instruments

The observation period was from 9 December 2014 to 22 April 2015. A storm occurred during the observation period. The sensors on the tripod failed to acquire all potential observation data as power malfunctions occurred. ASM-4 were only obtained for three days from 13–17 December 2014. The current meter recorded data for 57 days from 9 December 2014 to 5 February 2015. The turbidity meter obtained 91 days of data from 9 December 2014 to 11 March 2015, and 107 days of wave–tide gauge data were obtained from 9 December 2014 to 5 February 2015. Another observation was conducted in the same observational site and used the same instruments from 18 December 2016 to 11 January 2017, which obtained 25 complete days of data.

### 3.3. Calculation

The wave-induced shear stress was calculated by

$$\tau_{wmax} = 0.5\rho_w f_w U_{max}^2 \tag{1}$$

[52], where $\tau_{wmax}$ is the maximum wave-induced shear stress (Pa); $\rho_w$ is seawater density, assumed to be 1.025 g/cm3; $f_w$ is the wave friction coefficient, assumed to be 0.01 [14]; and $U_{max}$ is the maximum wave orbital velocity (m/s).

The current-induced shear stress was calculated as

$$\tau_c = C_d \rho_w U_c^2 / 10000 \tag{2}$$

[52] where $\tau_c$ is the current-induced shear stress (Pa); $C_d$ is the traction coefficient, which is assumed to be $3.1 \times 10^{-3}$ [14]; and $U_c$ is the observed current velocity (cm/s).

The bed shear stress due to the current when both currents and waves coexist is determined by

$$\tau = c_f \rho \left( U_c^2 + c_w U_{max}^2 \right)^{0.5} U_c \tag{3}$$

$$c_f = gn^2/h^{1/3} \tag{4}$$

where $n$ is Manning's roughness coefficient, and $c_w$ is a coefficient, assumed to be 0.65 [53].

In the research region, the direction of waves could be similar to that of currents as they are induced by wind. Therefore, the total shear stress is the sum of the wave- and current-induced shear stresses:

$$\tau = \tau_{wmax} + \tau_c \tag{5}$$

[54] where $\tau$ is the total stress (Pa), $\tau_{wmax}$ is the maximum wave-induced shear stress (Pa), and $\tau_c$ is the current-induced shear stress (Pa).

The critical shear stress ($\tau_{cri}$) is estimated as:

$$\tau_{cri} = 0.00364P^{1.47} + 0.047 \tag{6}$$

[55] where P is the cohesive force (KPa).

We employed measures for decomposing single-wide sediment loads [11,56] to analyse the source of the suspended sediment flux. The per-tidal-cycle current velocity, water depth, and suspended sediment content are decomposed into mean and pulsating values. The single-wide load can be expressed as:

$$\begin{aligned} T &= VCH = \left(\overline{V} + V'\right)\left(\overline{C} + C'\right)\left(\overline{H} + H'\right) \\ &= \overline{VCH} + V'C'\overline{H} + V'\overline{C}H' + V'\overline{CH} + \overline{V}CH' + \overline{V}C'\overline{H} + \overline{V}C'H' + V'C'H' \end{aligned} \tag{7}$$

[57] where T is the single-wide sediment load (mg/(cm*s)), C is the suspended sediment concentration (g/L), V is the current velocity (cm/s), and H is the water depth (cm). $\overline{V}$, $\overline{C}$, and $\overline{H}$ are the average values of the depth-averaged current velocity, water depth, and depth-averaged suspended sediment concentration, respectively, and $V'$, $C'$, $H'$ and are the pulsating values of the current velocity, water depth, and suspended sediment concentration, respectively. Over a single tide cycle, $\sum^{A'} = \sum^{C'} = \sum^{H'}$. The single-wide load during a tide cycle can be expressed as:

$$T = \overline{VCH} + V'C'\overline{H} + V'\overline{C}H' + \overline{V}C'H' + V'C'H' = T_1 + T_2 + T_3 + T_4 + T_5 \tag{8}$$

where C and H are positive constants, and V may be positive or negative. When V is positive, it is the flood tidal current velocity, and when V is negative, it is the ebb current velocity. $T_1$ and $T_3$ are the contributions of the advection transport volume and Stokes drift, and the sum of $T_1$ and $T_3$ is the transport volume. $T_2$, $T_4$, and $T_5$ are components of $C'$, and the sediment concentration in water can be approximated as a constant during a tide cycle. The values of $C'$ are affected by the two-way exchange of sediment between the sediment and water. The sum of $T_2$, $T_4$, and $T_5$ is the resuspended sediment volume.

The suspended sediment content can be expressed:

$$\sum_{i=1}^{H} C_i = CH, \tag{9}$$

where $C_i$ is the suspended sediment concentration at different depths (g/L). From Formulas (7) and (9), the resuspended sediment concentration can be expressed as:

$$T_2 + T_4 + T_5 = C_r VH \tag{10}$$

$$C_r VH/CVH = (T_2 + T_4 + T_5)/T \tag{11}$$

$$C_r = (T_2 + T_4 + T_5)\sum_{i=1}^{H} C_i/T \tag{12}$$

where $C_r$ is the resuspended sediment concentration (g/L). From the above formulas, the transport sediment concentration can be calculated as:

$$C_a = T_1 \sum_{i=1}^{H} C_i / T \tag{13}$$

$$C_s = T_3 \sum_{i=1}^{H} C_i / T \tag{14}$$

$$C_t = C_a + C_s \tag{15}$$

where $C_a$ is the advection transport sediment concentration (g/L), $C_s$ is the Stokes drift transport sediment concentration (g/L), and $C_t$ is the transport sediment concentration (g/L). Using the above formulas, the suspended sediment content and contribution of current transport (tidal current and Stokes drift) and resuspension to the suspended sediment content can be calculated.

## 4. Results

### 4.1. Hydrodynamic Parameters

The winter hydrodynamic parameters were measured for 25 days by taking *in-situ* observations. The observed hydrodynamic state variables are shown in Figures 3 and 4. During the first observation period, the site experienced a storm surge from 12:00 on 15 December to 06:00 on 17 December 2014. Before the storm, the observation site experienced an irregular half-day tide. The significant wave height was generally below 2.50 m, and the main sea conditions were moderate (significant wave height of 1.25–2.50 m). When the storm arrived, the significant wave height exceeded 3.0m, and the sea conditions became rough (significant wave height of 2.50–4.00 m). After the storm, the significant wave height decreased to below 2.50 m. The tide ranged from 0.52 to 2.92 m during the first observation period. The minimum and maximum water depths were 8.45 and 12.22 m, respectively. The maximum current velocity was 70.6 cm/s, and the maximum significant wave height of 4.03 m occurred at 24:00 on 15 December 2014 (Figure 3).

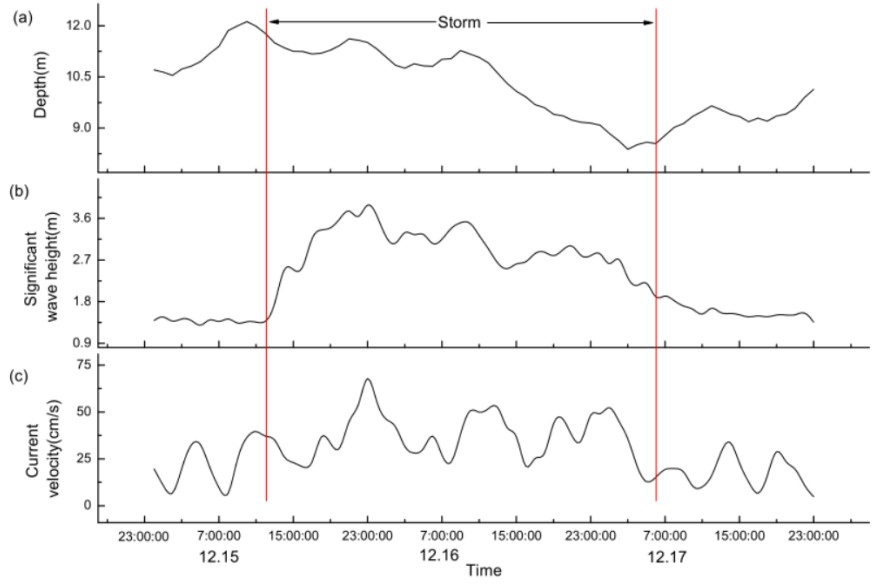

**Figure 3.** (**a**) Depth, (**b**) significant wave height, and (**c**) current velocity from 15–17 December 2014.

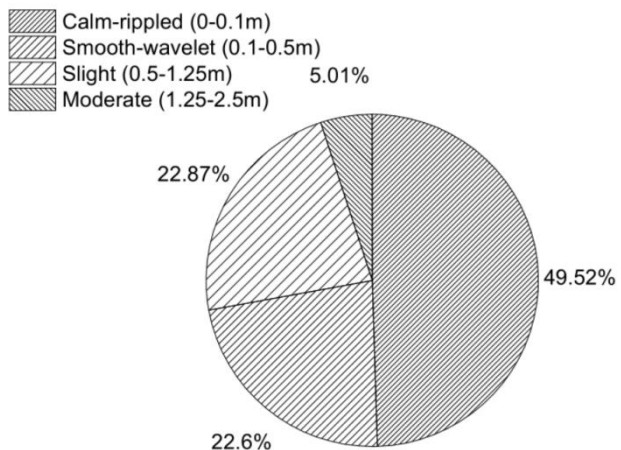

**Figure 4.** Percentage of the time period occupied by different sea conditions at the study site from 19 December 2016 to 11 January 2017.

During the second observation period, the sea conditions at the study site were mainly calm-rippled (significant wave height of 0–0.1 m) (Figure 4). However, on 22 and 26 December 2016, and 10 January 2017, the significant wave height increased by over 1 m due to wind. The tidal range during the second observation period was 0.12–1.21 m, and the minimum and maximum water depths were 6.62 and 8.93 m, respectively (Figure 5). The maximum of current velocity reached 49.40 cm/s. Generally, five sea conditions were observed, according to the in-situ measurements, including rough (2.5–4.0 m), moderate (1.25–2.5 m), slight (0.5–1.25 m), smooth-wavelet (0.1–0.5 m), and calm-rippled (0–0.1 m).

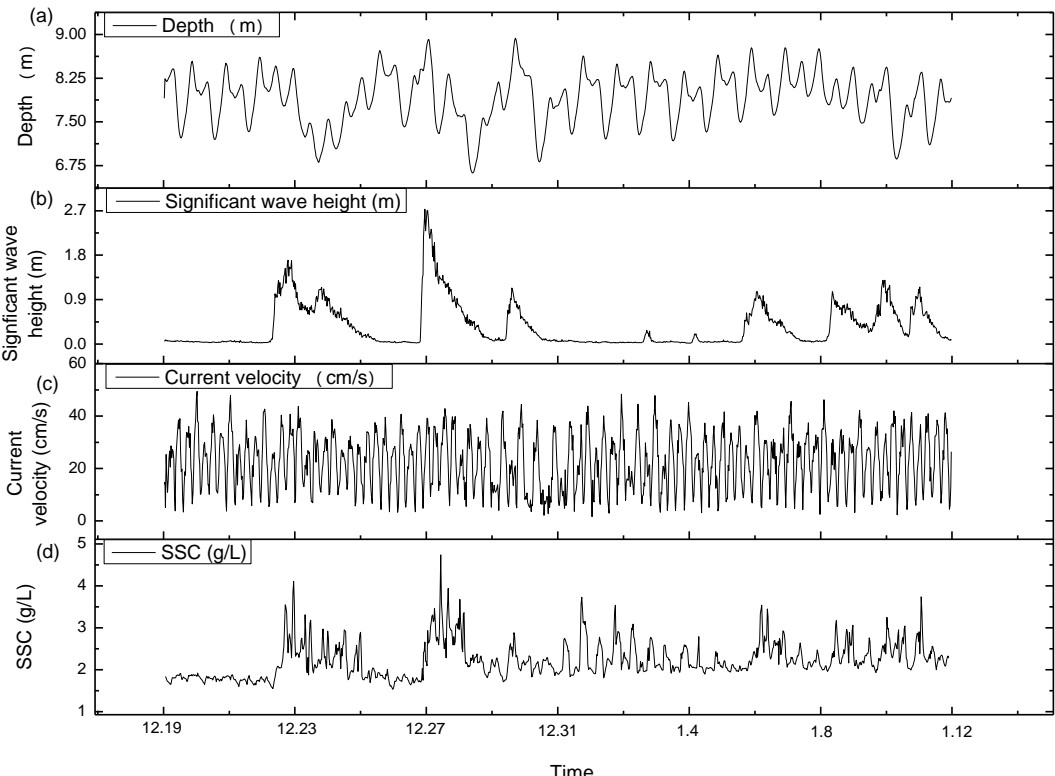

**Figure 5.** (**a**) Water depth, (**b**) significant wave height, (**c**) current velocity, and (**d**) average SSC from 19 December 2016 to 11 January 2017.

### 4.2. Suspended Sediment Concentration

The SSC varies greatly under different sea conditions. The SSC profiles show that during the observation period the SSC was mainly below 2.5 g/L, and its maximum level (>12.5 g/L) occurred during the storm surge (Figure 6). The high-tide period is normally characterized by increasing SSC, which exceeded 4.5 g/L. During the storm period from 15–17 December 2014, the SSC was well-distributed throughout the water column (Figure 6a), which could reflect the strong vertical mixing due to the enhanced waves or storm surge [58]. The SSC appeared to be irregular from 26 December 2016 to 4 January 2017 after the significant wave height increased to 2.7 m (Figures 5 and 6b). These irregular SSC (lutoclines) occurred at about 90cm above the seabed (Figure 6b), and these lutoclines resulted in average SSC records between 12/19 and 12/22 (all below 2 g/L), which is less than records between 12/30 and 01/05 (ranging from 2–3 g/L) despite the similar tide/current/wave conditions (Figure 5). After 5 January 2017, the SSC was also well-distributed in the water as the lutoclines were broken by stronger waves (Figure 6b). Thus, the hydrodynamic parameters under different sea conditions were the main factors affecting the vertical distributions of SSC.

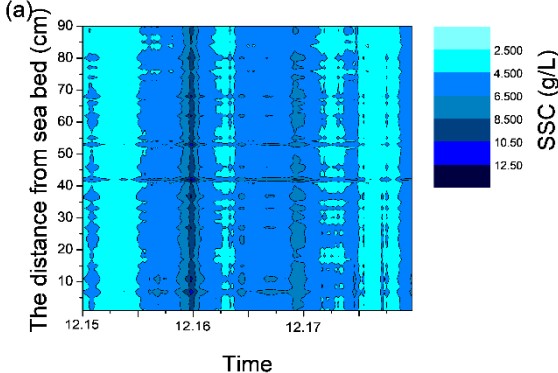

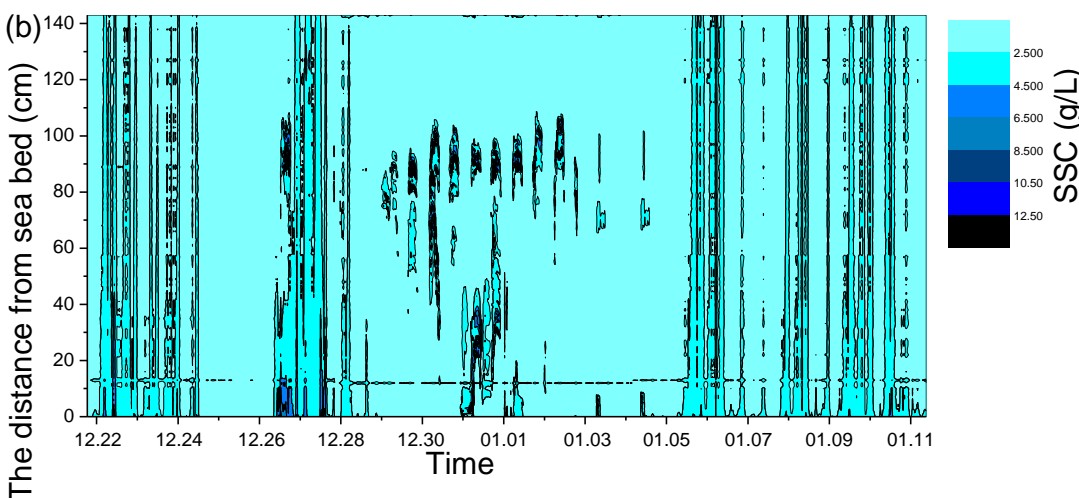

**Figure 6.** SSC, which was mainly less than 2.5 g/L, and its profiles during the periods of (**a**) 15–17 December 2014, (**b**) 21 December, 2016 to 11 January, 2017.

### 4.3. Physical and Mechanical Properties of Seabed Sediment

The 5 m sediment core showed that the sediment is mainly composed of silt (77.3%) and clay (18.5%) (Table 1). A new cone penetration test system (CPTs) [59], which was developed by the Ocean University of China, was used to test the in-situ sediment shear strength. The results show that there is a comparatively hard crust in the core from 0.8 to 1.6 m. Previous studies found that the sediment

strength in the study area is below 150 kPa [60], but our test results show a relatively high strength of 130–290 KPa (Table 2).

**Table 1.** Physical and mechanical properties of seabed soils collected in a 5 m sediment core.

| Physical Properties | | | | Mechanical Properties | |
|---|---|---|---|---|---|
| Water content (%) | 23.7 | Void ratio | 0.66 | Cohesive forces (kPa) | 18.99 |
| Wet density (g/cm$^3$) | 2.01 | Liquidity index | 0.6 | Internal angle of shearing resistance (°) | 24.34 |
| Degree of saturation (%) | 96.3 | Plasticity index | 7.23 | Coefficient of volume compressibility (MPa$^{-1}$) | 0.14 |

a. Data represent the mean values obtained from laboratory geotechnical tests following the standard soil testing method (GBT 50123-1999).

**Table 2.** Soil bearing capacity up to a depth of 2.5 m according to a CPT test.

| Depth (m) | 0–0.2 | 0.2–0.6 | 0.6–0.8 | 0.8–1.6 | 1.8–2.4 |
|---|---|---|---|---|---|
| Bearing capacity (KPa) | 130 | 210 | 170 | 290 | 220 |

a. Data represent the mean values obtained from the empirical formula $f_k = 0.043P_s + 0.06$ [61], where $f_k$ is the bearing capacity and $P_s$ is the cone resistance.

## 5. Discussion

### 5.1. Contribution of Waves and Currents to Resuspension

The suspended sediment is closely related to the sea conditions, which are, therefore, associated with the processes of sediment resuspension and transportation. During the observation period, the transport sediment concentration (TSC) accounted for 69.3%–100.0% of the SSC, and its maximum level was 2.8 g/L. (Figure 7).

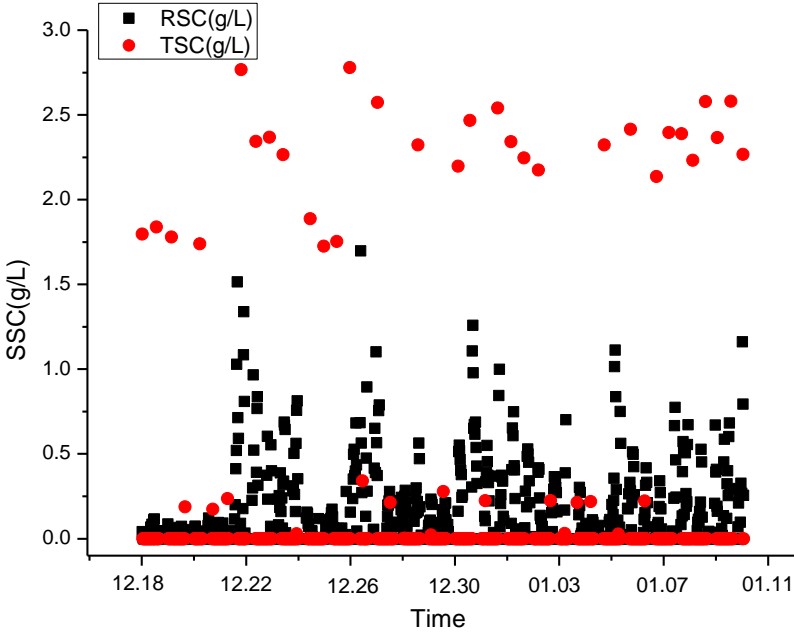

**Figure 7.** Resuspended sediment concentration (RSC) and transport sediment concentration (TSC) from 18 December 2016 to 11 January 2017.

Equation (5) was used to analyze the contributions of waves and currents, as the correlation between the SSC and total shear stress calculated by Equation (5) (Figure 8) was better than that calculated by Equation (3) (Figure 9). The R, R2 of Figure 8 are larger than in Figure 9, and the mean

square and standard errors of Figure 8 are less than Figure 9 (Table 3). During the first observation period, the study site experienced rough (significant wave height of 2.5–4.0 m) and moderate (significant wave height of 1.25–2.5 m) sea conditions. Under these two conditions, the fraction of the resuspended sediment concentration (RSC) induced by waves was much greater than that induced by currents. The contribution of waves to the total RSC was 85.1%–99.7% during the rough period (Figure 10e), and 71.4–99.6% during the moderate period (Figure 10d). The maximum RSC induced by waves reached 2.2 and 1.2 g/L, respectively.

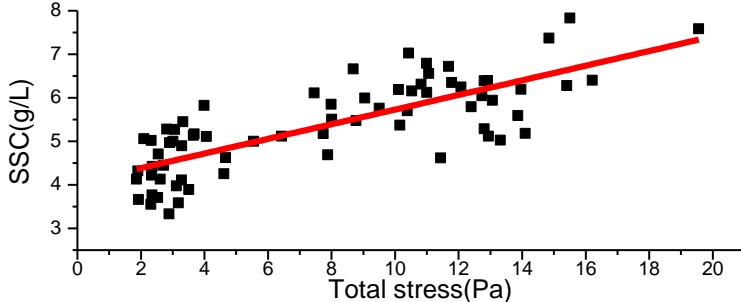

**Figure 8.** Suspended sediment concentration (SSC) as a function of the shear stress calculated by Equation (5) from 13–17 December 2014 (linear fit of the total stress Pearson's r is 0.78).

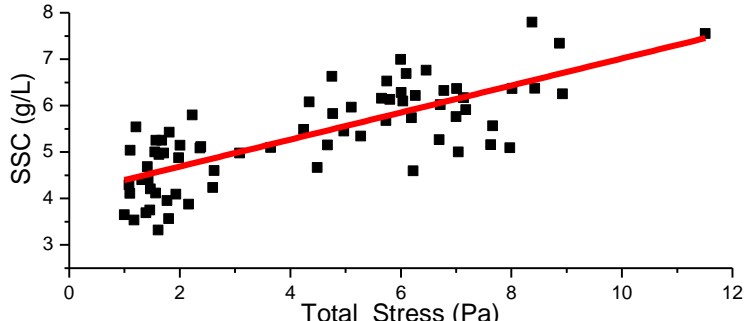

**Figure 9.** Suspended sediment concentration (SSC) as a function of the shear stress calculated by Equation (3) from 13–17 December 2014 (linear fit of the total stress Pearson's r is 0.75).

**Table 3.** The correlation and error of Figures 8 and 9.

| | R | $R^2$ | Mean Square Error | Intercept Standard Error | Slopestandard Error |
|---|---|---|---|---|---|
| Figure 8 | 0.78 | 0.60 | 0.42 | 0.15 | 0.016 |
| Figure 9 | 0.75 | 0.55 | 0.47 | 0.16 | 0.031 |

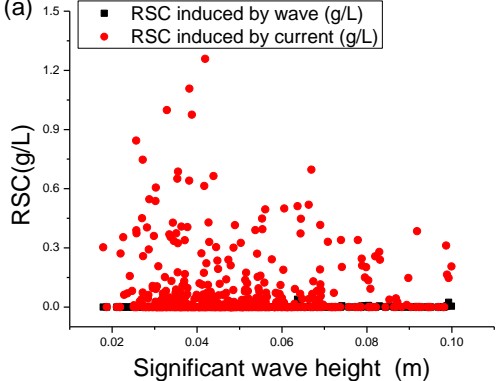

**Figure 10.** *Cont.*

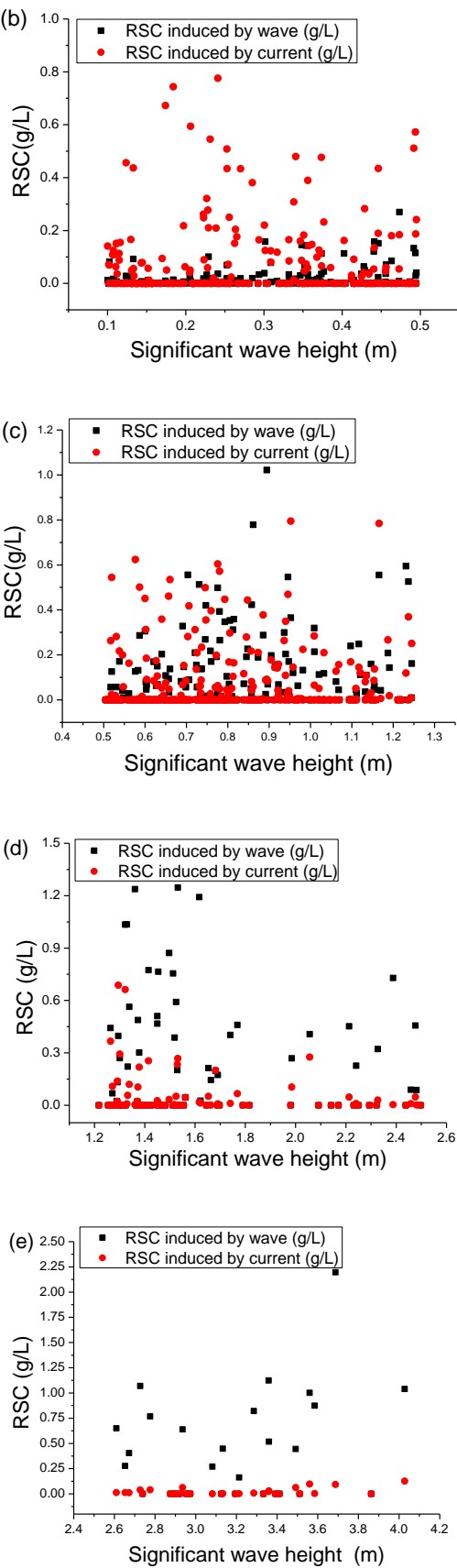

**Figure 10.** RSC induced by current or waves under (**a**) calm-rippled (significant wave height 0–0.1 m), (**b**) smooth-wavelet (significant wave height 0.1–0.5 m), (**c**) slight (significant wave height 0.5–1.25 m), (**d**) moderate (significant wave height 1.25–2.5 m), and (**e**) rough (significant wave height 2.5–4.0 m) sea conditions.

During the slight (significant wave height of 0.5–1.25 m) period, the RSC induced by currents was similar to that induced by waves (Figure 10c). The contribution of currents to the total RSC was between 3.0% and 87.8%, and the maximum RSC induced by currents and waves reached 0.79 and 1.02 g/L, respectively. Under smooth-wavelet (significant wave height 0.1–0.5 m) and calm-rippled (significant wave height 0–0.1 m) conditions, the RSC induced by currents was much higher than that induced by waves (Figure 10a,b). The fraction of RSC induced by currents during the smooth-wavelet and calm-rippled periods were as low as 30.9% and 77.7% of the total RSC, respectively, and the maximum RSC induced by currents under these two conditions reached 1.26 and 0.78 g/L, respectively.

Generally, the RSC induced by currents exceeded that induced by waves under calm-rippled and smooth-wavelet sea conditions, while the RSC induced by waves under slight, moderate, and rough sea conditions was higher than that induced by currents during the observation period. Although the RSC induced by currents decreased as the significant wave height increased when the sea conditions changed from calm-rippled to rough (Figure 10a–e), the current velocities changed slightly under different sea conditions (Figure 4). The result shows that the stress induced by currents positively affects resuspension when waves are weak. The effect of the current decreased when sediment resuspension increased with a significant increase in wave height.

## 5.2. Relationship between SSC and Resuspension

During the observation period, the SSC did not increase with a significant increase in the wave height beyond 1.25 m (Figure 11). From 26–27 December 2016, the total shear stress increased to 10 Pa as the significant wave height increased to 2.7 m, and then decreased rapidly (Figures 4b and 12). However, the SSC (including the averaged and near-bed SSCs) decreased slowly. The change in the near-bed SSC lagged behind the average SSC after the total stress reached 10 Pa (Figure 12). This indicates that the sediment does not immediately reach the seabed during flocculent settling. The flocculent settling changed to hindered settling when the SSC exceeded 3 g/L, and the hydrodynamic conditions did not distribute the SSC well in water. Under this condition, the settling velocity of suspended sediment decreased with increasing SSC [62]. This resulted in lutoclines with a thickness of approximately 20 cm (Figure 13). When the total shear stress exceeded the critical shear stress (0.32 Pa), which was calculated by Equation (6), the near bed SSC record between 17 December 2016 and 5 January 2017 was higher, despite the similar shear stress (Figure 14). This shows that the SSC exchange velocity between the lutoclines and lower water was reduced. The absolute value of SSC increased due to the decrease in settling velocity, and the SSC was higher in the lutoclines and lower water. When the significant wave height increased and broke the lutoclines, flocculent settling increased the settling velocity and caused the absolute SSC value to decrease. These effects weakened the resuspension-induced RSC, and the RSC induced by currents decreased when the significant wave height increased while the current velocities changed slightly.

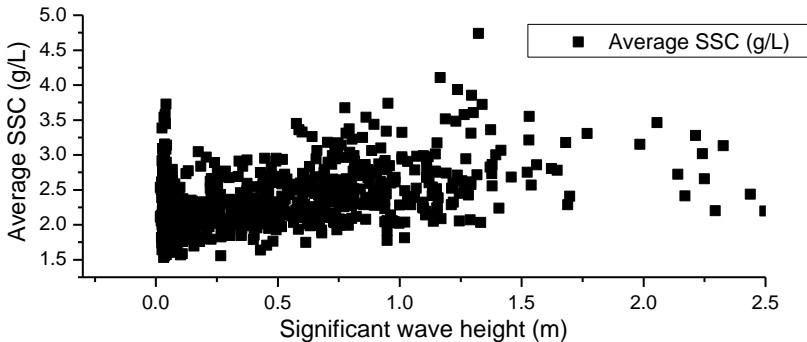

**Figure 11.** The average SSC under significant wave height below 1.25 m condition is not lower than the SSC when the significant wave height exceeds 1.5 m.

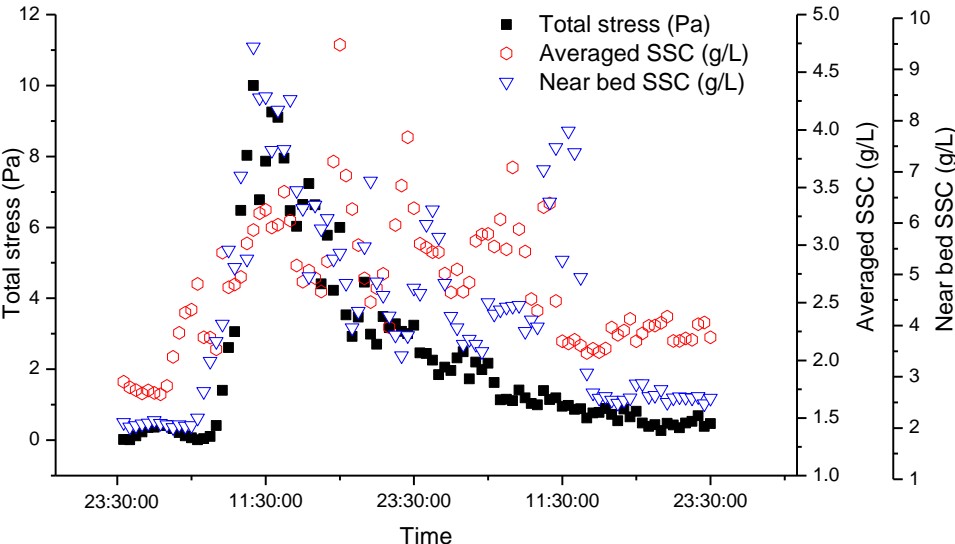

**Figure 12.** Changes in SSC and total stress with time during 26-27 December 2016.

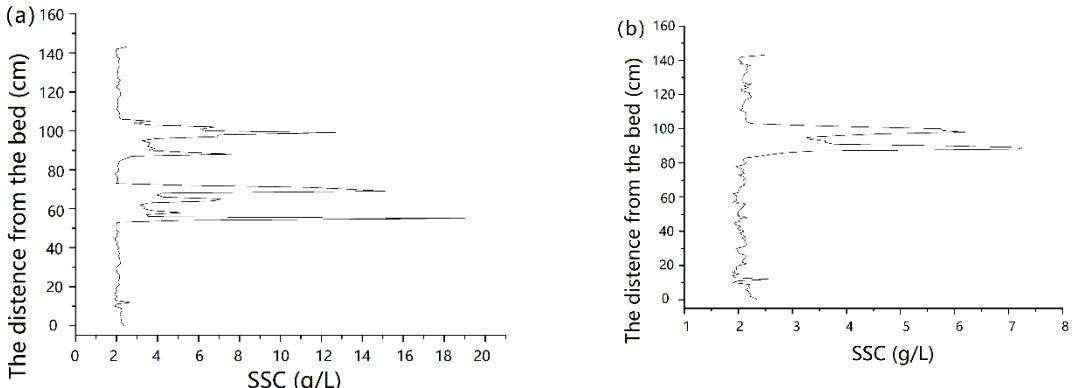

**Figure 13.** (**a**) The two lutoclines at 07:00 on 30 December 2016, when the maximum SSC exceeds 18 g/L; (**b**) a single lutocline at 10:30 on 1 January 2017.

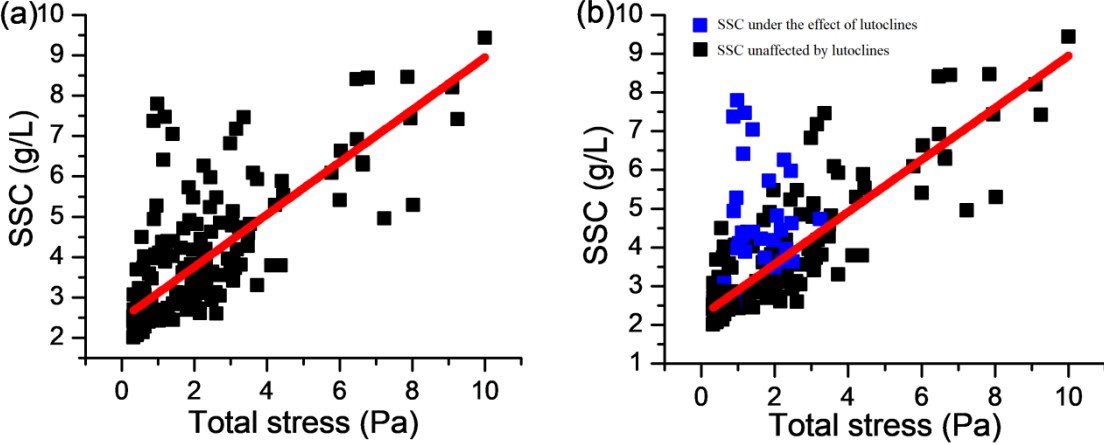

**Figure 14.** (**a**) The near bed SSC from 19 December 2016 to 12 January 2017 as a function of the shear stress calculated using Equation (6) (linear fit of the total stress Pearson's r is 0.75). (**b**) The near SSC from 19–26 December 2016 and from 6–13 January 2017 as a function of the shear stress calculated by Equation (6) (blue points are the SSC calculated by Equation (6) from 27 December 2016 to 5 January 2017. The linear fit of the total stress Pearson's r is 0.88).

## 6. Conclusions

The in-situ monitoring records show that the changes and sources of SSC changes mainly depend on the sea conditions in the Yellow River Delta-front. The SSCs were normally less than 2.5 g/L, and high SSCs (>4.5 g/L) occurred during high-tide periods. During the storm surge, the SSC reached its maximum (>12.5 g/L) and was uniform throughout the water column due to the strong vertical mixing caused by the enhanced waves. The main source of suspended sediment during the observation period is sediment transport, accounting for 69.3%–100.0% of the SSC. When the significant wave height is below 1.25 m, the contribution of currents to the total RSC exceeded that of waves. However, the contributions of waves to the total RSC were 85.1%–99.7% and 71.4%–99.6% during the rough and moderate periods, respectively. The 20 cm-thick lutoclines occurred after the significant wave height increased to its peak value and then decreased. When the flocculent settling of suspended sediment hindered in the Yellow River Delta when the SSC exceeded 3 g/L and the hydrodynamic conditions could not break the lutoclines. Under hindered settling, the resuspended sediment stagnated in the lutoclines and lower water as the settling velocity reduced from 27 December 2016 to 11 January 2017. When the significant wave height exceeded 1 m, the lutoclines were broken, the flocculent settling increased the settling velocity, and the SSC was well diffused in the water. This study reveals different controlling factors for the high SSC near a river-influenced delta, and helps us get a better understanding of a delta's resuspension and settling mechanisms. The future studies are comparing winter with summer.

**Author Contributions:** B.L. taked part in fieldwork, data analysis and paper writing; Y.J. designed the observation; J.P.L. revised the paper; X.L. and Z.W. taked part in fieldwork, and data analysis. All authors have read and agreed to the published version of the manuscript.

**Funding:** This research project was funded by the National Natural Science Foundation of China (grant No. 41072215, grant No. 41877221, grant No. 41877223, grant No. U1906230) Qingdao National Laboratory for Marine Science and Technology (grant No. QNLM2016ORP0110); and the Marine Geological Survey Project of the China Geological Survey (grant No. GZH201100203).

**Acknowledgments:** Special thanks go to the Ocean University of China for the joint training MSc programs and NC State University giving me a visiting scholar chance. Thanks go to Fudong Ji for their assistance with fieldwork. Anonymous reviewers valuable comments on a previous version of this paper.

**Conflicts of Interest:** The authors declare no conflict of interest.

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
