# Peer review of "Effect of Wave, Current, and Lutocline on Sediment Resuspension in Yellow River Delta-Front"

_water, doi:10.3390/w12030845_

Round 1

Reviewer 1 Report

The authors proposed an interesting topic presenting a SSC study in the Yellow River Delta from in-situ observations.
The work was thoroughly written, the authors carried out detailed analyzes, the results are very
interesting as well as the subject of the topic under consideration.
Only some little amendments are needed for a suitable paper improvement:

-the figures are all blurred, maybe you can fix it by increasing image resolution.
-please check in typos in text.

Author Response

Reviewer 1 Comments (Technical Comments to the Author): Comments: The work was thoroughly written, the authors carried out detailed analyzes, the results are very interesting as well as the subject of the topic under consideration.Only some little amendments are needed for a suitable paper improvement:

Response: Thanks for reviewing our manuscript and your recognition of our work. I have revised the paper following your comments.

Reviewer 1 Comments (Remarks to the Authors):

Comments: -the figures are all blurred, maybe you can fix it by increasing image resolution.

Response: Thank you for your suggestion, I have fixed some figures by increasing image resolution.

Comments: -please check in typos in text.

Response: Thank you for your suggestion, I have revised some typos that are marked by red in paper.

Reviewer 2 Report

The work presents an interesting method to understand the effect of wave, current, and lutocline on sediment resuspension in Yellow River delta-front.

The article is complete, but I suggest inserting some sentences in the introduction and conclusions to contextualize the topic. For example, the authors might mention that understanding sediment re-suspension mechanisms is useful for understanding the movement of pollutants. It's just a suggestion to improve the context but not mandatory.

Author Response

Reviewer 2 Comments (Technical Comments to the Author):

Comments:

The article is complete, but I suggest inserting some sentences in the introduction and conclusions to contextualize the topic. For example, the authors might mention that understanding sediment re-suspension mechanisms is useful for understanding the movement of pollutants. It's just a suggestion to improve the context but not mandatory.

Response:

Thanks for reviewing our manuscript and your recognition of our work. I will insert the sentences “And marine sediments serve as a key sink for heavy metals that are released into the sea when the seabed sediment is resuspended (Neto et al., 2000; Yuan et al., 2004).” in Line 40-41.

Reviewer 3 Report

This manuscript fits well with the journal. However, major revision is suggested for now. Data interpretation and presentation need to be improved before accept. The conclusions are not well supported by the data presented in this current version. In addition, language needs to be improved.

Line 18, hydrodynamic factors

Line 23, accounted

Line 25, were created

Line 30-32, Sentence needs to be reorganized.

Line 66, correlated to

Lines 128,135,137, do the authors mean precision? It seems that accuracy cannot be only 2%.

Line 129, and the sampling interval is 30 min.

Line 161, It looks like the authors use cm/s  a lot for current velocity. Would suggest the authors keep consistent on unit.

Line 209, the significant wave height exceeded 3m.

Line 219, not sure there is a term called effective wave height, and would suggest not to use it.

Figure 5, keep current velocity unit consistent with figure 3.

Figure 6, Keep y-axis the same range among a,b,c,d. X-axis has different formats among a, b, c, d, and color bar scales are different in a, b, c, d. Would suggest keep the format of x-axis consistent, such as using yy:dd:hh:mm for all of them, and keep color scales the same, which will be easy distinguish high-low SSC among figures.

Line 238-239, refer to the figures related (fig 5 and fig 6 (b-c)).

Line 239-242. It is difficult to understand, and suggest the authors reorganize this sentence. Besides, ‘these lutoclines’? the authors never constrain the lutoclines in the data or figure 6.

And also, it is really every hard to tell maximum ssc around 90 cmab (cm above bed) in figure 6(b-c).

Line268, what is transport sediment concentration?

Figure 7, keep the unit of ssc consistent with other figures and those in the text.

Line 270-272, why Figure 8 is better than Figure 9? R is very close. If the authors use R2, they should also be very close.

Line 276-278, how do the authors get those contribution? The figure 8 does clearly not show those contribution.

Line 278, mg/L or g/L? Please be consistent in the text and figures.

Line 279-287, Please refer to the related specific subplots in Figure 10 in this paragraph, and be consistent with the unit of SSC.

Line 299, Which subplot in Figure 10?

Line 318-320, This sentence does not read well, and suggest the authors reorganize it.

Line 321, Where is Figure 13?

Line321-324, Suggest the authors reorganize this sentence.

Line 339, it’s better to add a legend in figure 14.

Line 354, the data does not show a 20-cm lutoclines.

Line 360-361, Not sure distribute is the right word to use here. Suggest the authors change this word.

Line 362-363, grammar mistake in this sentence.

Author Response

Authors are honored that editors and reviewers gave the constructive comments on this manuscript. We followed the reviewers’ advice and revised the manuscript. All changes have been marked using the track changes mode in MS Word.

Reviewer 3 Comments (Technical Comments to the Author):

Comments:

This manuscript fits well with the journal. However, major revision is suggested for now. Data interpretation and presentation need to be improved before accept. The conclusions are not well supported by the data presented in this current version. In addition, language needs to be improved.

Response:

First of all, sincerely, thanks for reviewing our manuscript. It is our great honor to show the information what you are concerned.

Reviewer 3 Comments (Remarks to the Authors):

Comments:

Line 18, hydrodynamic factors

We followed the reviewer’s advice, revised it in Line 20, and reorganized the sentence. 

Comments: 

Line 23, accounted

Response:

We followed the reviewer’s advice and revised it in Line 28.

Comments:

Line 25, were created

Response:

We followed the reviewer’s advice and revised it in Line 29.

Comments:

Line 30-32, Sentence needs to be reorganized.

Response:

We followed the reviewer’s advice and reorganized the sentence in Line 34-36 as follows:

Original sentence:This study reveals the mechanisms the influencing formation of suspended sediment changes and distribution in the Yellow River Delta.

 Current sentence:This study reveals different controlling factors for the high SSC near a river-influenced delta, and helps us get a better understanding of a delta’s resuspension and settling mechanisms.

Comments:

Line 66, correlated to

Response:

We followed the reviewer’s advice and revised it in Line 71.

Comments:

Lines 128,135,137, do the authors mean precision? It seems that accuracy cannot be only 2%.

Response:

The accuracy means instrumental error.

Comments:

Line 129, and the sampling interval is 30 min.

Response:

We followed the reviewer’s advice and revised it in Line 136.

Comments:

Line 161, It looks like the authors use cm/s  a lot for current velocity. Would suggest the authors keep consistent on unit.

Response:

We followed the reviewer’s advice and changed the equation to keep consistent on unit.

Comments:

Line 209, the significant wave height exceeded 3m.

Response:

We followed the reviewer’s advice and revised it in Line 217.

Comments:

Line 219, not sure there is a term called effective wave height, and would suggest not to use it.

Response:

We followed the reviewer’s advice and changed this effective wave height to significant wave height.

Comments:

Figure 5, keep current velocity unit consistent with figure 3.

Response:

We followed the reviewer’s advice and changed the current velocity unit of Figure 5.

Comments:

Figure 6, Keep y-axis the same range among a,b,c,d. X-axis has different formats among a, b, c, d, and color bar scales are different in a, b, c, d. Would suggest keep the format of x-axis consistent, such as using yy:dd:hh:mm for all of them, and keep color scales the same, which will be easy distinguish high-low SSC among figures.

Response:

We followed the reviewer’s advice and changed the x-axis and colour scales of Figure 6 to keep the format consistent.  

Comments:

Line 238-239, refer to the figures related (fig 5 and fig 6 (b-c)).

Response:

We followed the reviewer’s advice and revised it in Line 246-247.

Comments:

Line 239-242. It is difficult to understand, and suggest the authors reorganize this sentence. Besides, ‘these lutoclines’? the authors never constrain the lutoclines in the data or figure 6. And also, it is really every hard to tell maximum ssc around 90 cmab (cm above bed) in figure 6(b-c).

Response:

We followed the reviewer’s advice and reorganized the sentence in Line 247-250 as follows:

Original sentence:The maximum SSC occurred at approximately 90 cm above the seabed (Figure 6 (b)), and these lutoclines resulted in average SSC records between 12/19 and 12/22 (all below 2 g/L) and between 12/30 and 01/05 (ranging from 2-3 g/L), despite the similar tide/current/wave conditions (Figure 5).

 Current sentence:These irregular SSC (lutoclines) occurred at about 90cm above the seabed (Figure 6 (b)), and these lutoclines resulted in average SSC records between 12/19 and 12/22 (all below 2 g/L) are less than records between 12/30 and 01/05 (ranging from 2-3 g/L), despite the similar tide/current/wave conditions (Figure 5).

Comments:

Line268, what is transport sediment concentration?

Response:

Transport sediment concentration means the concentration of suspended sediment transported by tidal current or residual current.

Comments:

Figure 7, keep the unit of ssc consistent with other figures and those in the text.

Response:

We followed the reviewer’s advice and changed the SSC unit to consistent with g/L.

Comments:

Line 270-272, why Figure 8 is better than Figure 9? R is very close. If the authors use R2, they should also be very close.

Response:

Because their R is very close,but the R, R2 of Figure 8 are more than Figure 9, and the mean square and standard errors of Figure 8 are less than Figure 9.

R

R2

Mean square error

Intercept standard error

Slope

standard error

Figure 8

Figure 9

0.79

0.75

0.60

0.55

0.42

0.47

0.15

0.16

0.016

0.031

Comments:

Line 276-278, how do the authors get those contribution? The figure 8 does clearly not show those contribution.

Response:

The contribution is got by the percentage of the RSC induced by wave or current to the total RSC.

Comments:

Line 278, mg/L or g/L? Please be consistent in the text and figures.

Response:

Thanks for your suggestion, the unit has been consistent with g/L.

Comments:

Line 279-287, Please refer to the related specific subplots in Figure 10 in this paragraph, and be consistent with the unit of SSC.

Response:

We followed the reviewer’s advice and revised it in Line 303-307.

Comments:

Line 299, Which subplot in Figure 10?

Response:

The subplots is Figure 10 (a-e) in Line 320-322.

Comments:

Line 318-320, This sentence does not read well, and suggest the authors reorganize it.

Response:

We followed the reviewer’s advice and reorganized the sentence in Line 333-336 as follows:

Original sentence:This is because, when the SSC exceeded 3 g/L, and as the hydrodynamic conditions cannot distribute the SSC well in water, the settling velocity decreased with increasing SSC (Figure 13) under hindered settling (Ross, 1988).

Current sentence:The flocculent settling changed to  hindered settling when  the SSC exceeded 3 g/L, and the hydrodynamic conditions did not distribute the SSC well in water. Under this condition, the settling velocity of suspended sediment decreased with increasing SSC (Ross, 1988).

Comments:

Line 321, Where is Figure 13?

Response:

Sorry, I have reinserted Figure 13.

Comments:

Line321-324, Suggest the authors reorganize this sentence.

Response:

We followed the reviewer’s advice and reorganized the sentence in Line 337-339 as follows:

Original sentence: When the total shear stress exceeded the critical shear stress (0.32 Pa), which was calculated by Eq. (6), the near bed SSC from 27 December 2016 to 3 January 2017 exceeded that during the other time periods (Fig. 14) experiencing similar total stress.

 Current sentence:When the total shear stress exceeded the critical shear stress (0.32 Pa), which was calculated by Eq. (6), the near bed SSC record between 12/17/2016 and 01/03/2017 was higher, despite the similar shear stress (Figure 14).

Comments:

Line 339, it’s better to add a legend in figure 14.

Response:

Thanks for your suggestion, I have added a legend in Figure 14.

Comments:

Line 354, the data does not show a 20-cm lutoclines.

Response:

Figure 13 shows the 20-cm lutoclines.

Comments:

Line 360-361, Not sure distribute is the right word to use here. Suggest the authors change this word.

Response:

I change distribute to diffuse in Line 374. Do you think this word is better?

Comments:

Line 362-363, grammar mistake in this sentence.

Response:

We followed the reviewer’s advice and reorganized the sentence in Line 375-378 as follows:

Original sentence:This study reveals the mechanisms the influencing formation of suspended sediment changes and distribution in the Yellow River Delta; future comparison studies are compare with summer season are needed.

Current sentence:This study reveals different controlling factors for the high SSC near a river-influenced delta, and helps us get a better understanding of a delta’s resuspension and settling mechanisms. The future comparison studies are compared winter season with summer season.
